# Synthesis of Iron Nanomaterials for Environmental Applications from Hydrometallurgical Liquors

**Christiana Mystrioti, Nymphodora Papassiopi and Anthimos Xenidis \***

School of Mining and Metallurgical Engineering, National Technical University of Athens, 15780 Athens, Greece; chmistrioti@metal.ntua.gr (C.M.); papasiop@metal.ntua.gr (N.P.)

\* Correspondence: axen@metal.ntua.gr

**Abstract:** Hydrometallurgical leaching solutions are often rich in iron, which was traditionally considered a major impurity. However, iron can be selectively separated and recovered by applying appropriate solvent extraction and stripping techniques, and the resulting solutions can be valorized for the synthesis of high-added-value products, such as magnetic iron oxide nanoparticles (mIONPs). The aim of this study was to synthesize high-quality mIONPs from solutions simulating the composition of two alternative stripping processes. The precursor solutions consisted of Fe(II) in an acidic sulfate environment and Fe(III) in an acidic chloride medium. The Fe(II)-SO$_4$ solution was treated with a mixture of KNO$_3$-KOH reagents, and the product (M(II)) was identified as pure magnetite with a high specific magnetization of 95 emu·g$^{-1}$. The treatment of Fe(III)-Cl solution involved the partial reduction of Fe(III) using metallic iron and the co-precipitation of iron cations with base addition combined with microwave-assisted heating. The product (M(III)) was a powder, which consisted of two phases, e.g., maghemite (75%) and magnetite (25%), and also had a high magnetic saturation of 80 emu·g$^{-1}$. The nanopowders were evaluated for their effectiveness in removing Cr(VI) from contaminated waters. The maximum adsorption capacity was found to be equal to 11.4 and 17.4 mg/g for M(II) and M(III), respectively. The magnetic nanopowders could be easily separated from treated waters, a property that makes them promising materials for the water treatment sector.

**Keywords:** magnetite; maghemite; nanoparticles; mIONPs; pregnant leaching solution (PLS) valorization; adsorption capacity; Cr(VI) removal; water treatment

## 1. Introduction

Magnetic iron oxide nanoparticles (mIONPs), such as magnetite (Fe$_3$O$_4$) and maghemite (γ-Fe$_2$O$_3$), are ideal for a wide variety of applications due to their magnetic properties. Magnetite and maghemite have a similar cubic spinel structure. In the case of magnetite, the tetrahedral sites are occupied by Fe(III) ions, and the octahedral sites are evenly filled by Fe(II) and Fe(III), while in maghemite, all the iron cations are in the trivalent state. The two oxides possess ferrimagnetic characteristics, and it is difficult to distinguish between them on the basis of their magnetic properties at room temperature [1]. The saturation magnetization (Msat) of bulk magnetite is around 90–100 emu·g$^{-1}$, and the Msat of maghemite ranges from 74 to 80 emu·g$^{-1}$, while for hematite, it is only 0.1–0.4 emu·g$^{-1}$ at room temperature [2].

The magnetic properties and the relatively simple and low-cost synthesis procedure of magnetite and maghemite nanoparticles have led to numerous applications in the biomedical, technological, nanohydrometallurgical and environmental sectors. The biomedical applications include magnetic resonance imaging (MRI), targeted drug delivery, hyperthermia of tumors for cancer therapy, etc. [3,4]. Iron oxide nanoparticles are preferred to other nanoparticles (lanthanides or manganese), which are also used for MRI because they are associated with fewer neurotoxicity symptoms. It has also been reported that they can enhance metabolic activities in patients due to the accumulation of iron in human organs [5,6].

The technological applications include several electronic and magnetic storage devices, ferrofluids, magneto-optic sensors, alternative anodes in lithium batteries, etc. [2,7].

In the environmental sector, iron oxide nanoparticles also have strong potential. Their physical properties, such as small particle size (nanosize) and large specific surface area (ranging from 30 to 100 $m^2/g$), make them ideal as low-cost and effective adsorbents for water treatment. It should be mentioned that magnetite and maghemite have lower specific surfaces compared to other iron oxyhydroxides, such as amorphous ferrihydrite (100–700 $m^2/g$). However, their main advantage is that they can be separated from treated waters very rapidly and efficiently by applying simple magnetic separation techniques. The effectiveness of these mIONPs for the removal of cationic heavy metals, anionic contaminants such as Cr(VI) or organic compounds has been investigated by several researchers for either magnetite [8–14] or maghemite [15]. It is noted that magnetite combines reductive and adsorbing properties, whereas maghemite functions only as an adsorbent.

Magnetic iron nanoparticles have been investigated in the last decade in the innovative field of nanohydrometallurgy. Supermagnetic magnetite was applied as a magnetic agent for capturing rare earths ($La^{3+}$ and $Nd^{3+}$) from solutions via complexation reactions. The effective capture of lanthanide ions using iron oxide nanoparticles functionalized with ethylenediaminepropylsilane and diethylenediaminepentaacetic acid agents was evaluated in some studies [16,17]. The separation of iron nanoparticles was more than 99% and was achieved after three successive complexation and release stages [17].

Magnetic iron nanoparticles could be synthesized using iron from the pregnant leaching solution (PLS) derived from several hydrometallurgical processes, such as the electrolytic zinc process or the hydrometallurgical treatment of low-grade laterites containing very high concentrations of iron, which must be separated from metals of commercial value, e.g., Ni and Zn. Iron is thus considered a major impurity, and the development of appropriate iron control processes is a continuous struggle for hydrometallurgists [18]. Several purification techniques, such as precipitation, solvent extraction and ion exchange [19,20], have been studied for iron separation from valuable metals such as Cu, Ni and Co.

Until 1960, the only known method of removing iron was to raise the pH of the acidic leach solution to a pH of approximately 3 in order to cause the precipitation of iron as ferric hydroxide. The resulting precipitate was a voluminous, amorphous material that occluded a great part of the mother solution, and the subsequent processes of filtering and washing this gelatinous residue presented major difficulties. Due to these drawbacks, the method of hydroxide precipitation was limited to process solutions containing no more than 1–2 g/L iron [18]. In the mid-1960s, several processes were developed and commercialized involving the precipitation of iron in the form of crystalline and easily filterable iron compounds, such as jarosite, goethite and hematite. In the jarosite process, the precipitation of iron is achieved by adjusting the pH to 1.5 at a temperature of approximately 95 °C. This process was widely applied by several zinc companies in Spain, Australia and Norway [18,19]. The goethite process was developed and commercialized in Belgium [20]. In this process, the iron is initially reduced to the divalent state, followed by oxidation with air at a temperature of approximately 90 °C and at a pH controlled at approximately 3.0. The process requires very fine control of pH and of the whole oxidation procedure, because the ferric iron concentration should not rise above 1 g/L. The hematite process was developed and applied in Japan and in Germany and requires high-temperature and high-pressure conditions, i.e., 200 °C and 2 MP [21].

In all of the above processes, the produced iron oxyhydroxides are mainly discarded in dumps. The massive recycling of iron to other industrial sectors could not be achieved. An example is the Ruhr Zink company (Frankfurt, Germany), which implemented the hematite process in order to sell the iron oxide to steelmakers. This aim was not obtained due to the included Zn and S impurities. According to [18], in order to achieve the required purities, as specified by the steelmaking or pigment sectors, the best available technical solution is solvent extraction.

Solvent extraction, as an alternative for the selective separation of iron from polymetallic solutions, has been widely investigated since the mid-1980s [22]. The extractants that have been used cover a wide range of organic compounds, including carboxylic acids [23], organophosphorus compounds [24], amides [25], alcohols and ketones [26]. Many of the above extractants were found to be successful in the selective extraction of Fe from hydrometallurgical liquors; however, the stripping of iron from the loaded organic phase proved to be a difficult task and has limited the development of similar technologies to full commercial scale. Ion exchange using chelating resins has been effectively applied for selectively metal recovery. The presence of ferric iron in the solution decreased the performance of a chelating resin due to its selectivity order. A pre-reducing step of ferric iron to ferrous by sodium dithionite was added in order to enhance the metal recovery [27,28].

In a previous work, our team studied the separation of Fe from the pregnant leaching solution produced during the hydrometallurgical treatment of low-grade laterites [29]. The iron was selectively extracted from the PLS by using a mixture of di-2-ethylhexyl phosphoric acid (D2EHPA) and tri-butyl phosphate (TBP) extractants. The recovery of Fe from the loaded organic solution was also studied by evaluating two methods: (a) acidic stripping, using $H_2SO_4$ solutions, and (b) reductive (galvanic) stripping, using elemental iron, Fe(0), as a reductant [30]. The best results, namely, 90% iron recovery, were obtained by applying galvanic stripping. In this process, the loaded organic was mixed with a 0.25 M sulfuric acid solution in the presence of metallic iron. The elemental iron caused the reduction of Fe(III) to the divalent state and facilitated the transfer of Fe(II) to the aqueous phase. Alternative stripping processes for the removal of Fe(III) from D2EHPA organic phases are based on the use of highly acidic HCl solutions. For instance, the stripping of Fe from a D2EHPA–TBP mixture was obtained with a 2 M HCl solution [31], while Fe stripping with D2EHPA as a single extractant was obtained with even more acidic solutions of 3 to 8 M HCl [31].

The aim of this study was to valorize the iron from the pregnant leaching solution for the synthesis of mIONPs with magnetic properties. To the best of our knowledge, there has been no attempt until now to use iron-rich hydrometallurgical solutions as the primary source for the synthesis of magnetic iron nanoparticles. All previous studies dealing with the synthesis of mIONPs used commercially available ferric and ferrous salts. The specific novelty of the present study is the effort to synthesize magnetic iron nanoparticles from solutions simulating the composition of the above-mentioned stripping processes and evaluate how the different compositions of precursor solutions affect the properties and the effectiveness of nanoparticles for use in environmental applications. Specifically, the starting solutions consisted of Fe(II) in a sulfate medium, as derived from the galvanic stripping process, or Fe(III) in a chloride medium, as produced from the HCl stripping alternative. The precipitation of magnetic nanoparticles from the Fe(II)-$SO_4$ solution was carried out by applying a single-step procedure, using a mixture of $KNO_3$ and KOH for the partial oxidation of Fe(II) and the formation of magnetite. The synthesis of mIONPs from the Fe(III)-Cl solution was carried out by applying a novel process, which consisted of two steps. In the majority of published studies, the precipitation of nanomagnetite from chloride solutions was carried out by mixing a ferrous chloride and a ferric chloride salt at the required molar ratio of 1 to 2 [3,5–7]. In this work, the required amount of ferrous iron was produced by the controlled partial reduction of ferric iron to the divalent state using metallic iron [20]. The next step consisted of the co-precipitation of Fe(II) and Fe(III) by the addition of ammonia, followed by a microwave heating treatment.

The physicochemical and the magnetic properties of the two types of nanoparticles were compared, and their adsorption capacity for the treatment of polluted waters with Cr(VI) was evaluated. The objective of this study was to explore the possibility of transforming iron from a disturbing impurity in hydrometallurgical liquors into a valuable source for the synthesis of novel marketable products, such as the mIONPs presented in this study, which can decrease the environmental footprint of the mining industry and contribute to the circular economy.

## 2. Materials and Methods

### 2.1. Materials

Ferric chloride ($FeCl_3 \cdot 6H_2O$), aqueous ammonia 18 M (28–30% $w/w$) and absolute ethyl alcohol were purchased from Sigma Aldrich and were used for the preparation of iron oxide nanoparticles from Fe(III)-Cl solutions. Metallic iron ($H_2Omet$ 86) was supplied by RioTinto Metal Powders LTD (Sorel-Tracy, QC, Canada). $H_2Omet$ 86 is a coarse, high-density iron powder with particle sizes less than 250 μm and bulk density equal to 3.28 g/cm$^3$.

$FeSO_4 \cdot 7H_2O$, $KNO_3$ and KOH were also supplied by Sigma Aldrich (St. Louis, MO, USA) and were used for the synthesis of magnetite from Fe(II)-$SO_4$ solutions. Potassium dichromate ($K_2Cr_2O_7$) and 1,5 diphenylcarbazide were obtained from Merck, Darmstadt, Germany, and were used for chromium removal experiments and for the analysis of Cr(VI), respectively.

### 2.2. Synthesis of Magnetic Iron Nanoparticles

Two different procedures were followed for the synthesis of magnetic nanoiron, depending on the composition of the precursor solution, which was prepared to simulate the aqueous phase emanating from the two alternative stripping processes.

The solution simulating galvanic stripping contained 0.1 M $FeSO_4$ and 0.15 M $H_2SO_4$ and was prepared using deionized water, which had been previously purged with $N_2$ at 90 °C to remove dissolved oxygen. The synthesis was carried out by applying a variation of Sugimoto and Matijevic's (1980) procedure [32]. Specifically, 500 mL of the Fe(II) solution was heated at 90 °C under a $N_2$ atmosphere in a Pyrex beaker 1 L. When the temperature was reached, 50 mL of a solution containing 6 M KOH and 1.1 M $KNO_3$ was added with a flowrate of 5 mL/min using a peristaltic pump (Watson Marlow Alitea 313DSI, Stocklholm, Sweden). The suspension was heated for 60 min and then left to cool overnight. Added KOH was in stoichiometric excess with respect to the amount required for the neutralization of $H_2SO_4$ and the precipitation of Fe(II) as $Fe(OH)_2$. The concentration of hydroxyls in the final solution was close to 0.09 M. According to Sugimoto and Matijevic, the first step in this process is the formation of ferrous hydroxide gel, $Fe(OH)_2$, which is then oxidized by nitrate anions to form magnetite crystals. The nitrates are gradually reduced to lower oxidation states and are finally transformed into ammonia. The overall stoichiometry can be described by reaction (1):

$$12\,Fe(OH)_2 + NO_3^- + H^+ \rightarrow 4\,Fe_3O_4 + NH_3 + 11\,H_2O \tag{1}$$

The added amount of $KNO_3$ was determined in order to have an initial concentration of of 0.1 M in the reactive mixture. In the same study [32], it was reported that the optimum concentration of $KNO_3$ was close to 0.1 M. At lower concentrations, the particle growth slowed down, but higher levels of $KNO_3$ did not accelerate the process.

The solution simulating the HCl stripping of Fe(III) from D2HEPA-TBP mixtures contained 0.2 M $FeCl_3$ and 1.4 M HCl. The solution was treated with elemental iron using conical tubes in order to obtain the partial reduction of Fe(III) to the Fe(II) state according to the stoichiometry of reaction (2):

$$Fe(0) + 2Fe(III) \rightarrow 3Fe(II) \tag{2}$$

A preliminary experiment was conducted to study the kinetics of reduction of ferric iron to ferrous iron when Fe(0) is added to the solution at a dose of 1.75 g/L. This dose corresponds to a stoichiometric excess of 25% regarding the desired ratio Fe(II)/Fe(III) = 1/2 M/M in the final solution. The results of the test are shown in Figure 1. The ferrous iron concentration was determined by redox titration using potassium permanganate, and Fe(tot) was analyzed using AAS. Fe(III) was calculated. It was found that the desired ratio can be obtained within 2 h. The above procedure was used to prepare 1 L of a stock solution containing 0.15 M Fe(III) and 0.075 M Fe(II), which was kept under a nitrogen atmosphere.

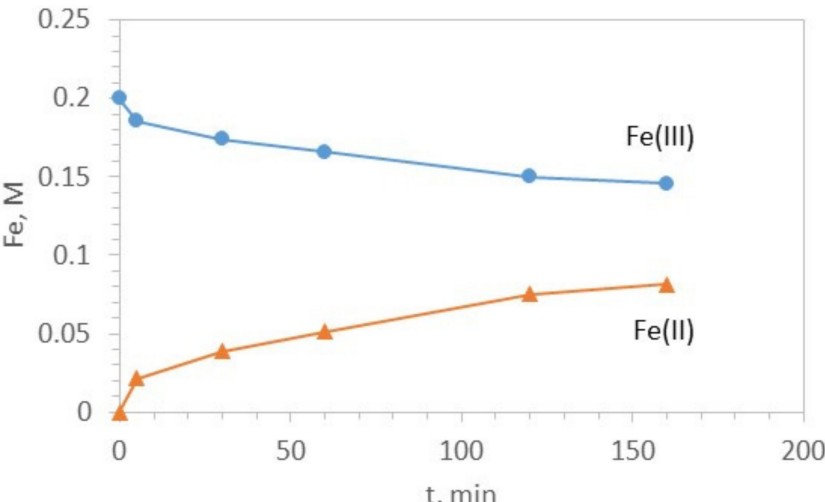

**Figure 1.** Kinetics of Fe(III) reduction with Fe(0). Initial solution: 0.2 M FeCl$_3$ and 1.4 M HCl. Fe(0) dose 1.75 g/L.

The synthesis of nanomagnetite from this precursor solution was carried out in batches of 40 mL. The solution was heated at 80–90 °C, and ammonia solution, with a concentration of 18 M, was added under continuous stirring until the pH of the solution was equal to 9. The color of the suspension changed from yellow to black with the increase in pH. The suspension was then placed in a microwave synthesis lab station (Microsynth from Milestone, (Sorisole, Italy) for a heating time of 90 s, applying vigorous stirring by software-controlled magnetic stirrer and microwave power of 160 W.

Microwave-assisted heating provides fast reaction kinetics and homogeneous heat supply due to the direct contact of the electromagnetic field with magnetite [33].

Following their synthesis, the iron oxide nanoparticles were separated from the aqueous solution using a neodymium magnet and were washed twice with deionized water and ethanol alcohol. Then, the nanoparticles were dried in an oven at 45 °C.

### 2.3. Characterization of Nanoparticles

The nanomagnetic powders were examined by X-ray diffraction. The analysis was performed using a Bruker D8-Focus powder diffractometer (Bruker, Karlsruhe, Germany) with nickel-filtered CuKa radiation ($\lambda$ = 1.5405 Å). Morphological analysis of the nanopowders was performed by TEM analysis (JEM 2100 HR, Jeol, Tokyo, Japan) operated at 200 kV. The BET (Brunner–Emmett–Teller) specific surface area of the nanopowders was determined by N$_2$ adsorption at −196 °C using a NOVA 1200 gas sorption analyzer (Quantachrome Instruments, Boynton Beach, FL, USA). TG-DTA/DSC analysis was carried out using the Labsys 1200 apparatus under He atmosphere (Setaram Inc., Lyon, France). The samples were heated from 22 to 700 °C at a constant rate of 10 °C min$^{-1}$. Mössbauer spectra and magnetization saturation (Msat) were determined at NCSR "Demokritos". Mössbauer spectroscopy was carried out in constant acceleration mode with a Co$^{57}$ (Rh) source. Msat was determined using a superconducting quantum interface device-vibrating sample magnetometer (Quantum Design SQUID magnetometer Darmstadt, Darmstadt, Germany). The point of zero charge (pHpzc) was determined in selected samples by the acid–base potentiometric titration method [23]. In this method, a series of suspensions was prepared by mixing 0.25 g of the nanopowder with 50 mL of NaCl (0.01 M). The initial pH was adjusted to values between 2 and 12 with the addition of HCl or NaOH, and equilibrium pH was measured after 24 h of mixing.

### 2.4. Cr(VI) Removal Experiments

The adsorption isotherms were determined by implementing batch tests with initial Cr(VI) concentrations: 2, 5, 10, 20, 40, 60, 80 and 100 mg/L, and constant adsorbent dose

of 1 g/L. The experiments were carried out in duplicate using shaking flasks, which were placed in an orbital agitator. The suspensions were agitated at 250 rpm for 240 min, and the temperature was kept constant at 25 °C. Samples were analyzed for Cr(VI) using the USEPA 7196a method in a DR-1900 spectrophotometer (HACH, Loveland, CO, USA) with detection limit equal to 15 μg/L. The experiments were carried out in Cr(VI) solutions at natural pH, which ranged between 4.5–5.0 and remained approximately the same during the adsorption experiments. The separation of nanoparticles took place easily using a magnet.

## 3. Results

### 3.1. Characterization of Nanomagnetic Iron Oxides

3.1.1. X-ray Diffraction Analysis (XRD)

The XRD patterns of nanoparticles are presented in Figure 2. The nanomagnetic particles produced from the Fe(II)-$SO_4$ solution simulating galvanic stripping are denoted as M(II), while those synthesized from Fe(III)-Cl solutions are denoted as M(III). Six characteristic peaks ((220), (311), (400), (422), (511) and (440)) can be seen in the XRD patterns of both samples. It should be noted that it is not possible to distinguish between magnetite ($Fe_3O_4$) and maghemite ($\gamma$-$Fe_2O_3$) from the XRD analysis, because the two minerals have very similar crystallographic characteristics.

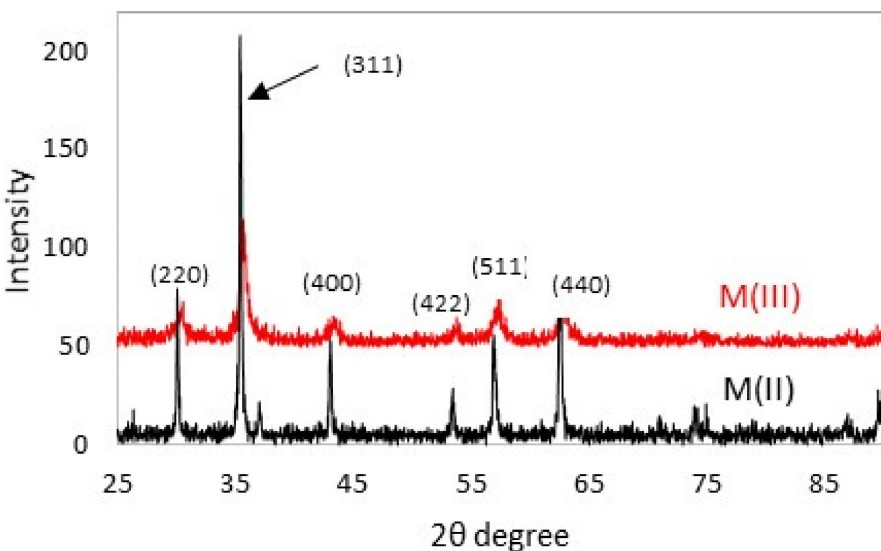

**Figure 2.** XRD pattern of nanoparticles.

The pattern of M(II) presents high-intensity and sharp peaks, while that of M(III) has broad and low-intensity peaks, which are characteristic of materials with small particle sizes. The mean crystallite size was determined from XRD data using the Debye–Scherrer equation:

$$D = ((k \times \lambda))/((\beta \times \cos\theta)) \tag{3}$$

where k = constant, $\lambda$ = X-ray wavelength and $\beta$ = full width at half maximum (FWHM). The FWHM was calculated using Origin Software. The average crystallite size of the sample prepared by partial oxidation of Fe(II) was calculated to be about 35 nm, while that produced by partial reduction of Fe(III) was close to 11 nm.

3.1.2. Morphology and Specific Surface

TEM images of samples M(II) and M(III) are shown in Figure 3. As seen in Figure 3a, sample M(II) consists mainly of cubic particles with sizes varying between 20 and 100 nm. Some hexagonal particles are also clearly distinguished in the sample. The formation of cubic particles with comparable particle size distributions has been reported by Sugimoto and Matijevic [32] and Schwertmann and Cornel [34], who applied a similar synthesis

procedure. Sample M(III) consists of spherical particles with a rather homogenous size close to 10–15 nm (Figure 3b).

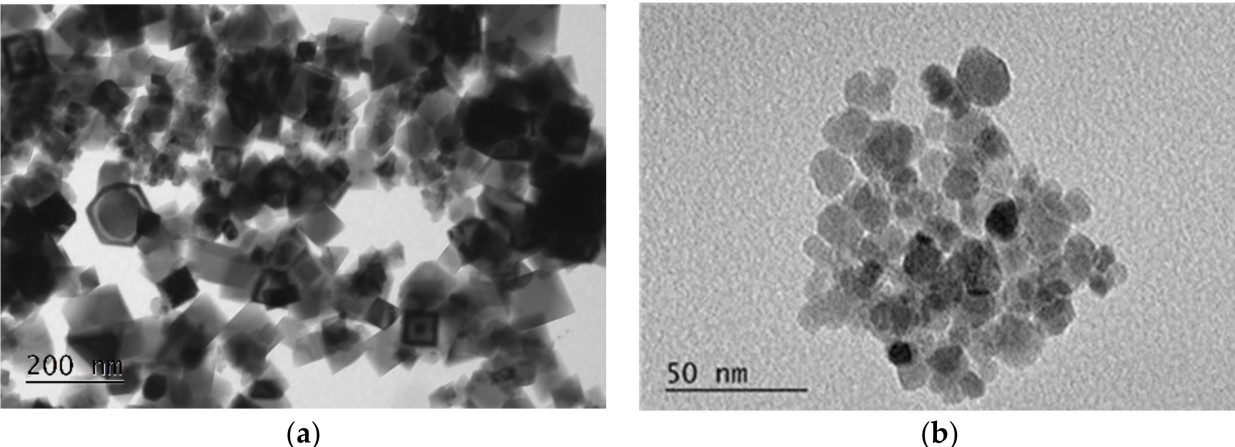

(**a**)　　　　　　　　　　　　　　　　(**b**)

**Figure 3.** TEM images of (**a**) M(II) and (**b**) M(III) nanoparticles.

The specific surface area, as determined by the BET method, was equal to 17.0 m$^2$/g for sample M(II) and 88.7 m$^2$/g for sample M(III).

### 3.1.3. Magnetic Properties

The Mössbauer spectra of the two samples are shown in Figure 4. For the M(II) sample, the room-temperature spectrum was typical of bulk-like magnetite. The spectrum was fitted using two components corresponding to Fe$^{+3}$ ions at site A and (Fe$^{+3}$ Fe$^{+2}$) ions on site B, with intensity area ratio B:A $\simeq$ 1:1. These characteristics are related to the electron transfer process between Fe$^{+2}$ and Fe$^{+3}$ on the octahedral B site and take place at temperatures above 115–150 K (Verwey temperature) [35].

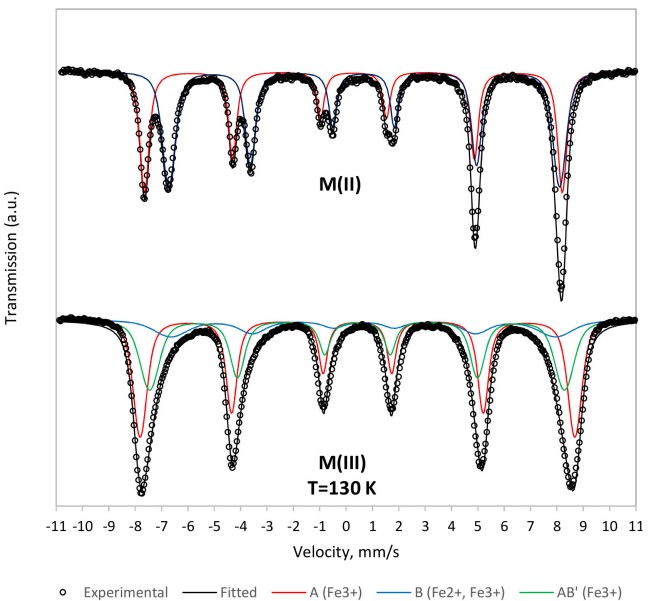

**Figure 4.** Mössbauer spectra of the two magnetic iron nanoparticle samples.

In the M(III) spectrum, the distinct fields of A and B components merged into a single sextet. The analysis was carried out at a lower temperature, i.e., 130 K, in order to improve peak resolution, but the main characteristics of the spectrum did not change.

The spectrum of M(III) was fitted with three components. The first component (A) corresponded to Fe$^{+3}$ ions in a tetrahedral environment, and its intensity was close to 46.4%.

The second component describing the contribution of $Fe^{+2}$, $Fe^{+3}$ in the octahedral B sites of magnetite was found to have a low intensity in the order of 16.2%. The third component (B') with 37.4% intensity was necessary to describe the total spectrum. The third component gave a signal for $Fe^{+3}$ in a high-spin state and can be attributed to maghemite, $\gamma$-$Fe_2O_3$.

From the relative contribution of the three components, it can be calculated that sample M(III) is a solid solution of magnetite (25%) and maghemite (75%). Pankratov et al. (2019, 2020) described similar compounds with the general formula $Fe_3$-$\delta O_4$, where $\delta$ can be derived from Mössbauer parameters and provides a measure of the oxidation of $Fe^{+2}$ [36,37]. In the case of sample M(III), $\delta$ is close to 0.26 and corresponds to a high degree of $Fe^{+2}$ oxidation, i.e., approximately 80% [36,37]. The saturated mass magnetization (Msat) of the two samples was determined using a vibrating sample magnetometer at a temperature of 5 K. For the magnetite sample, M(II), the Msat was equal to 95 emu·g$^{-1}$, while for the maghemite rich sample, M(III), Msat was slightly lower, i.e., 80 emu·g$^{-1}$.

### 3.1.4. Thermal Analysis

The TG and DSC curves of the magnetite nanoparticles are presented in Figure 5a,b. The TG curve of M(II) nanoparticles (Figure 5a) indicates a weight loss in the order of 8%, which takes place between 25 and 300 °C. This can be attributed to the loss of physically and chemically bound water, as well as the desorption of loosely bound anions, such as $SO_4$ and $NO_3$, originating from the precursor synthesis solutions. Thermal analysis of iron nitrate and sulfate compounds indicates that under an inert atmosphere, the removal of nitrates takes place in the temperature range between 100 and 140 °C [38], while that of sulfates is extended to higher temperatures up to 700 °C [39,40]. The DSC curve exhibits a wide endothermic peak between 25 and 170 °C and a small one at 310–330 °C. A clear exothermic peak was recorded at 535 °C and can be attributed to the formation of hematite ($\alpha$-$Fe_2O_3$) by the following equation:

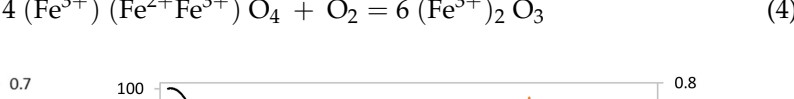

$$4\ (Fe^{3+})\ (Fe^{2+}Fe^{3+})\ O_4\ +\ O_2 = 6\ (Fe^{3+})_2\ O_3 \tag{4}$$

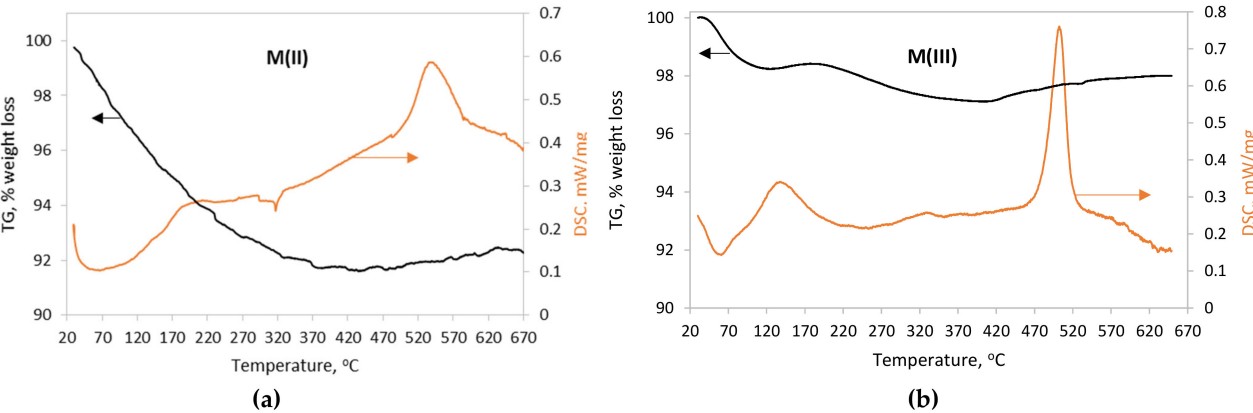

**Figure 5.** TG/DSC patterns of nanoiron particles (**a**) M(II) and (**b**) M(III).

Usually, pure samples of magnetite are not transformed into hematite at this temperature when the thermal analysis is carried out under an inert atmosphere [41]. The appearance of the peak suggests the presence of maghemite traces, despite the fact that it was not detected in the Mössbauer analysis of sample M(II).

For the M(III) nanoparticles, the weight loss (~2%), accompanied by an endothermic phenomenon at the temperature range from 25 to 150 °C, was attributed to the loss of physically absorbed water. The additional weight loss (~3%) between 150 and 500 °C is probably related to a small amount of chemically bound water. The heat flow presents a strong exothermic peak at 500 °C, which is characteristic of the transformation of maghemite ($\gamma$-$Fe_2O_3$) into hematite ($\alpha$-$Fe_2O_3$). This is in agreement with the transition temperature

reported by Darezereshki (2011) [42], who studied $\gamma$-Fe$_2$O$_3$ nanoparticles of similar size (13 nm).

### 3.1.5. Point of Zero Charge

The pH$_{pzc}$ of magnetite samples was determined by plotting the graph of initial pH vs. final pH values. The point of zero charge is the pH value at which the electric charge on the adsorbent surface is 0. The pH point of zero charge is given by the intersection of the curve with the straight line. The pH$_{pzc}$ values were found to be equal to 6.1 and 7.3 for samples of M(III) and M(II), respectively. The pH$_{pzc}$ suggests that the electric charge of the nanomagnetite surface changes from positive to negative values below and above this pH value. The pH is a crucial parameter for the adsorption of chromates on magnetic iron oxide nanoparticles.

The main properties of M(II) and M(III) nanoparticles are presented in Table 1.

**Table 1.** Main properties of synthesized magnetic iron oxide nanoparticles.

| Sample | Composition (Mössbauer) | Crystallite Size (XRD) nm | Particle Size (TEM) nm | SSA * m²/g | Msat emu·g⁻¹ | pH$_{pzc}$ |
|---|---|---|---|---|---|---|
| M(II) | Fe$_3$O$_4$ (100%) | 35 | 20–100 | 17.0 | 95 | 7.3 |
| M(III) | Fe$_3$O$_4$ (25%), $\gamma$-Fe$_2$O$_3$ (75%) | 11 | 10–15 | 88.7 | 80 | 6.1 |

* Specific Surface area.

### 3.2. Chromate Adsorption Results

The two types of magnetic nanoparticles were tested for their ability to adsorb Cr(VI). The adsorbed amounts of Cr(VI) on the solid phase (q$_e$, mg/g) as a function of the residual concentration in the aqueous phase (C$_e$, mg/L) are presented in Figure 6.

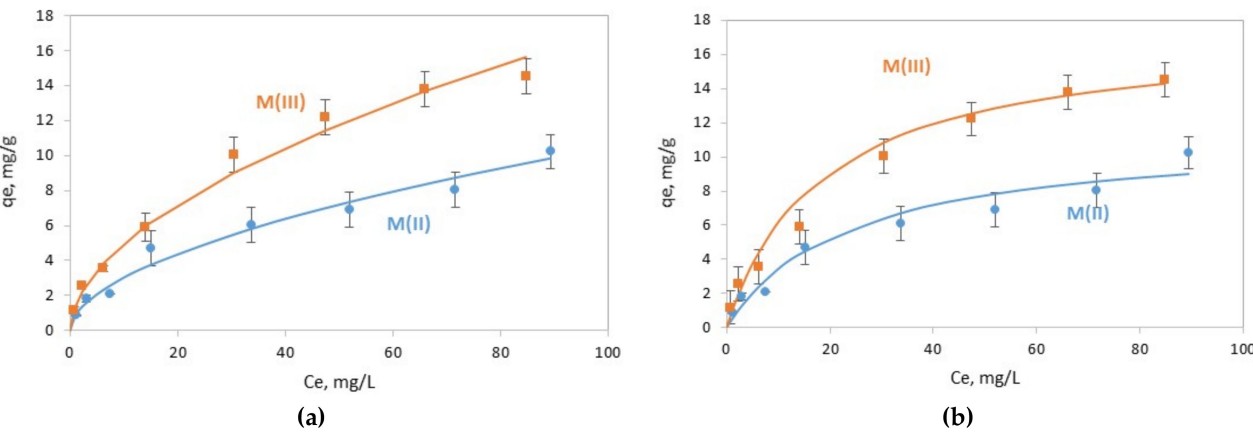

**Figure 6.** Adsorption of C(VI) by M(II) and M(III) nanoparticles. Fitting curves with (**a**) Freundlich and (**b**) Langmuir models.

The Freundlich and Langmuir models were applied in order to examine if they can describe the trend of Cr(VI) removal. The equations of the two models and the parameters that were determined by fitting the models to the experimental data are presented in Table 2. The Freundlich isotherm model exhibited slightly better fitting results, with coefficient correlations $R^2$ of 0.992 and 0.976 for M(III) and M(II), respectively, than the Langmuir model ($R^2 > 0.95$ for M(III) and $R^2 > 0.93$ for M(II)) for the two types of magnetite (Table 2). According to the Langmuir model, the maximum Cr(VI) adsorption capacity was equal to 11.4 mg/g and 17.4 mg/g for M(II) and M(III), respectively.

**Table 2.** Fitted parameters of Langmuir and Freundlich Models.

| Sample | Freundlich Isotherm: $q_e = K_F C_e^{1/n}$ | | |
|---|---|---|---|
| | n | $K_F$ (mg$^{(1-1/n)}$ L$^{(1/n)}$ g$^{-1}$) | $R^2$ |
| M(II) | 1.85 | 0.871 | 0.976 |
| M(III) | 1.87 | 1.451 | 0.992 |
| Sample | Langmuir Isotherm: $q_e = q_{max} K_L C_e / (1 + K_L C_e)$ | | |
| | $q_{max}$ (mg/g) | $K_L$ (L/mg) | $R^2$ |
| M(II) | 11.4 | 0.043 | 0.928 |
| M(III) | 17.4 | 0.054 | 0.968 |

The performance of the two magnetic iron nanoparticles regarding Cr(VI) is compared with other published data in Table 3. The adsorption experiments in these studies were carried out at various pHs. $HCrO_4^-$ anions mainly exist at a solution pH from 2.0 to 6.5. The higher content of $H^+$ in the solution results in the protonation of the mIONP surface, where the negatively charged $HCrO_4^-$ can be adsorbed by electrostatic adsorption. At higher pH values of the solution, the $CrO_4^{2-}$ species are prevalent, and the negatively charged magnetite is electrostatically repulsed. For comparison reasons, the adsorption capacity presented in Table 3 corresponds to results recorded in the pH range 4–5. According to most studies in Table 3, the adsorption capacity of Cr(VI) ranges from 3 to 14 mg/g, which was comparable to the results of the present study. Only Lasheen et al. (2014) demonstrated one order of magnitude higher adsorption capacity, namely, 121 mg/g, in comparison with the other published studies. This high adsorption capacity was estimated based on the Langmuir isotherm, and the magnetite used was very fine, with particle sizes from 2 to 7 nm [11].

**Table 3.** Comparison of M(II) and M(III) performance for Cr(VI) removal with published data.

| Mionp Type | Particle Size (nm) | Adsorption Capacity at pH 4–5 (mg/g) | Reference |
|---|---|---|---|
| Magnetite | 10 | 14 | [43] |
| Maghemite | 30 | 4 | [44] |
| Mixture magnetite, maghemite | 20–40 | 6 | [45] |
| Humic acid-Magnetite | 15 | 3.4 | [46] |
| Magnetite | 2–7 | 121 | [11] |
| Magnetite (MII) | 35 | 11.4 | Present study |
| Mixture magnetite, maghemite (M(III)) | 11 | 17.4 | Present Study |

It should be mentioned that information on the magnetic properties of mIONPs is rarely given. Hu et al. 2005 only mentioned that the Msat of maghemite particles was found to be equal to 3.3 emu·g$^{-1}$, indicating low magnetic properties [43,44]. In the present study, the prepared magnetite M(II) and the mIONP particles, M(III), possess high magnetic properties, which lead to effective magnetic separation and make them ideal for the treatment of polluted waters.

Various adsorbents have been evaluated for the removal capacity of chromates from polluted waters. Activated carbon (AC) and nanometals supported on carbon materials are promising mediums with adsorption capacities ranging from 13 to 120 mg/g [47]. However, the high cost of AC preparation inhibits their wide use. Clay materials are one of the most important groups of natural and abundant adsorbents for chromate removal. Adsorption

capacities of kaolin were tested and found to range from 10 to 195 mg/g. The adsorption of Cr(VI) in montmorillonite was lower than 6 mg/g [47].

## 4. Discussion

Hydrometallurgical solutions often contain elevated iron concentrations, which can be valorized as iron precursors for the synthesis of high-added-value materials such as magnetite and other mIONP powders. Iron can be selectively removed from polymetallic pregnant solutions by applying solvent extraction techniques and finally recovered without other admixtures in the stripping solutions. D2EHPA is an extractant with high selectivity for Fe(III), but the stripping process, which requires breaking the strong binding forces of the Fe(III)-D2EHPA complex, is a difficult task. In a previous work [29], we demonstrated that an efficient stripping process consists of reducing Fe(III) to the divalent state. The binding of ferrous iron with D2EHPA is much weaker, and it is thus easier to obtain the transfer of iron in the aqueous phase. This was achieved using elemental Fe(0) as a reductant and a 0.25 M $H_2SO_4$ solution as the stripping medium. According to other published data [31], it is possible to strip Fe(III) from D2EHPA-TBP mixtures using a 2 M HCl solution.

In this study, we tried to synthesize magnetic iron nanoparticles from solutions simulating the composition of the above stripping solutions. The starting solutions consisted of Fe(II) in a sulfate medium or Fe(III) in a chloride medium.

The magnetite powder from the Fe(II)-$SO_4$ solution was synthesized in a single-step procedure by adding a mixture of $KNO_3$ and KOH. The Mössbauer analysis of the sample indicated that the material consisted of pure magnetite. The mean particle size, as estimated by TEM data, was equal to 60 nm. This procedure for the synthesis of nanomagnetite was initially applied by Sugimoto and Matijevic [32]. The researchers found that the size, shape and composition of the produced nanoparticles were strongly affected by the type of iron precursors ($FeSO_4$, $Fe(NH_4)_2(SO_4)_2$, $FeCl_2$ and $Fe(CH_3CO_2)_2$), namely, by the coexisting anions. With ferrous acetate as the starting material, it was not possible to produce magnetite using standard synthesis conditions. The $FeCl_2$ solutions yielded very fine magnetite particles. With $FeSO_4$ and $Fe(NH_4)_2(SO_4)_2$ salts, it was possible to produce magnetite with a broad range of modal particle diameters, depending on the Fe(II)/OH molar ratio. When Fe(II) was in excess, the mean particle size was high, in the order of 400–800 nm, while in the presence of excess $OH^-$, the mean particle size was much finer, in the order of 10–50 nm. The researchers concluded that sulfate anions play a special and favorable role in the formation of magnetite. Our results confirm the findings of Sugimoto and Matijevic [32]. The nanoparticles produced from the Fe(II)-$SO_4$ solution presented a clear magnetite Mössbauer spectrum, and the XRD pattern suggested a material of high crystallinity. Moreover, the nanoparticles were found to have strong magnetic properties, with specific magnetization almost equivalent to that of bulk magnetite, i.e., 95 emu·g$^{-1}$ at 5 K.

The preparation of mIONPs using Fe(III)-Cl solution involved two steps: the partial reduction of ferric iron to the divalent state using metallic iron and the co-precipitation of Fe(II) and Fe(III) by the addition of ammonia, followed by a microwave heating treatment. When applying this method, the produced nanopowder had a fine average particle size (11 nm), and the Mössbauer analysis indicated that it consisted of two phases, namely, 75% maghemite and 25% magnetite. The specific saturated magnetization was slightly lower compared to that of sample M(II), i.e., 80 emu·g$^{-1}$ at 5 K.

It is noted that Msat depends on several parameters, such as the composition of nanoparticles, magnetite or maghemite, the particle size and also the temperature. As far as temperature is concerned, many studies have shown that Msat at 300 K is 10–20 emu·g$^{-1}$ lower compared to Msat at 5 K [3,35]. It is also known that magnetite has higher magnetic properties compared to maghemite. According to Shokrollahi [2], bulk maghemite has Msat of 74–80 emu·g$^{-1}$ at room temperature, while bulk magnetite has 92 emu·g$^{-1}$. The particle size finally has a strong effect on Msat. Goya et al. [35] synthesized nanomagnetite samples

with average particle sizes from 4 to 150 nm. Msat at 300 K was equal to 32 emu·g$^{-1}$ at 4 nm and increased up to 76 emu·g$^{-1}$ at 150 nm. At 5 K, the corresponding Msat ranged from 56 up to 89 emu·g$^{-1}$. The ultrafine particles with d = 4 nm produced by Goya et al. were also found to exhibit superparamagnetic properties at room temperature; i.e., they did not maintain any magnetization at zero external magnetic field.

The magnetic nanoparticles produced in this study were evaluated for environmental applications and specifically for their efficiency as an adsorption substrate for the removal of metal pollutants from contaminated waters. The experiments were carried out using Cr(VI) as a typical contaminant. The adsorption capacity for chromium removal was found to be equal to 17.4 mg/g using M(III) and 11.4 mg/g using M(II). This can be attributed to the higher surface area of M(III) in comparison with that of M(II). Despite the lower adsorption capacity of M(II), it has some important advantages, such as stronger magnetic properties and lower susceptibility to oxidation. M(III) was more sensitive to oxidation and should be stored under nitrogen, while magnetite, M(II), was resistant to oxidation, and there was no need for extra protection.

In addition to the field of environmental technologies, magnetic iron nanoparticles also have a strong potential for penetration in many other market sectors. However, certain applications are very demanding regarding the required properties of magnetic nanoparticles. For instance, in order to promote mIONPs in the biomedical field, it is important to synthesize nanoparticles combining high purity, strong magnetic properties and specific characteristics, such as hydrophilicity and biocompatibility. The two latter properties can be achieved by the appropriate functionalization of their surfaces using organic polymers such as dextran, PEG, starch, etc. [6]. This is a field of intensive ongoing research worldwide, which can lead to the production of several innovative and high-added-value products in the future.

## 5. Conclusions

In this study, ferrous and ferric iron solutions simulating the iron recovered from different stripping processes were evaluated as precursors for the synthesis of magnetic iron nanoparticles. It was found that the nanoparticles produced from the Fe(II)-SO$_4$ solution, M(II), consisted of pure nanomagnetite, nFe$_3$O$_4$, with an average particle size of 60 nm. The nanoiron oxides produced from the Fe(III)-Cl solution, M(III), consisted of maghemite $\gamma$-Fe$_2$O$_3$ (75%) and magnetite Fe$_3$O$_4$ (25%) with an average particle size equal to 11 nm. Their effectiveness in removing Cr(VI) from contaminated waters was equal to 11.4 and 17.4 mg/g for M(II) and M(III), respectively. Both products had strong magnetic properties, 95 and 80 emu·g$^{-1}$, which allowed rapid and efficient separation of the loaded nanoparticles from treated waters.

**Author Contributions:** Conceptualization, N.P.; methodology, N.P. and A.X.; validation, C.M.; formal analysis, C.M.; investigation, C.M.; resources, C.M.; data curation, A.X.; writing—original draft preparation, C.M.; writing—review and editing, C.M.; visualization, C.M.; supervision, N.P. and A.X.; project administration, C.M.; funding acquisition, C.M. All authors have read and agreed to the published version of the manuscript.

**Funding:** This research was co-funded by Greece and the European Union (European Social Fund (ESF)) through the Operational Program "HUMAN RESOURCES DEVELOPMENT, EDUCATION AND LIFELONG LEARNING" in the context of the project "Reinforcement of Postdoctoral Researchers—2nd Cycle" (MIS-5033021), implemented by the State Scholarships Foundation (IKΥ).

**Data Availability Statement:** MDPI Research Data Policies.

**Acknowledgments:** The authors would like to thank Eamonn Devlin for the Mössbauer and VSM analyses.

**Conflicts of Interest:** The authors declare no conflict of interest. The funders had no role in the design of the study; in the collection, analyses or interpretation of data; in the writing of the manuscript; or in the decision to publish the results.

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
