# Peer review of "Synthesis of Iron Nanomaterials for Environmental Applications from Hydrometallurgical Liquors"

_minerals, doi:10.3390/min12050556_

Round 1
Reviewer 1 Report
The paper deals about an interesting topic. However, authors need to highly the novelty and need to perform a major discussion of the traditional technologies used for the removal of ferric/ferrous ions in the leaching solutions.
I have follow questions:
Lines 56-59 "The effectiveness of these mIONPs for the removal of cationic heavy metals, anionic contaminants like Cr(VI) or organic compounds, has been investigated by several researchers for either magnetite [8-14] or maghemite [15]. It is noted that magnetite combines reductive and adsorbing properties, whereas maghemite functions only as adsorbent". Authors need to perform a wide discussion of the novelty and need of the proposed research.
Lines 60-62. Authors said "The magnetic iron nanoparticles can be synthesized using iron from the pregnant leaching solution (PLS) derived from several hydrometallurgical processes, such as the electrolytic zinc process or the hydrometallurgical treatment of low grade laterites". There are references of this or are proposed by authors?
Line 66. Please, indicate the name of PLS
Lines 69-70. "The recovery of Fe from the loaded organic phase was obtained applying a process known as “galvanic” stripping" Please, indicate reference. Why not direct stripping? Authors need to perform a better discussion of the presence of iron in leach liquors, separation methods, e.g. forming jarosites, stripping and washing in solvent extraction processes,
Quality of Figures can be improved
Characterization of iron oxide nanoparticles: I recommend the characterization by SEM or N2 isotherms
Conclusions needs to be revised. In this section authors need to highly the main conclusions of the research and not paragraph of the Introduction
Reviewer 2 Report
The present work aimed the production of magnetic iron oxide nanoparticles from hydrometallurgical processes. The material was further tested for chromium removal. The topic is important and will attach attention from scientific society.
The nanohydrometallurgy has been studied and should be presented and discussed.
A few questions are addressed to the authors. For instance, synthetic solutions were prepared for experiments, and I asked if the authors tested real solutions and if it will impact the quality of nanoparticles.
- What unit is “emu/g”?
- Page 1 line 33: what is “F(III)”? Is it fluorine?
- Page 2 lines 66-68: Other studies tried to solve the problem of iron in solution by several strategies, including ion exchange resins (see https://www.hindawi.com/journals/ijce/2018/9161323/, https://onlinelibrary.wiley.com/doi/full/10.1002/cjce.23306, https://onlinelibrary.wiley.com/doi/full/10.1002/cjce.23359, https://www.tandfonline.com/doi/full/10.1080/01496395.2019.1574828; https://doi.org/10.1016/j.jmrt.2019.07.059). Please, state this approach as a part of the literature review.
- How was the presence of ferrous iron controlled in the process?
- Figure 2 – please, indicate in the diffractogram the peaks and respective compounds.
- Adsorption data: were the experiments performed in duplicate? what is the experimental error?
- Compare adsorption data with current techniques. What is the removal efficiency?
- standardizes the units: use “emu/g” (as in Introduction) or “emu.g-1” (as in Conclusion).
Reviewer 3 Report
Manuscript ID: minerals-1671775
Title: Synthesis of iron nanomaterials for environmental applications from hydrometallurgical liquors
Authors: Christiana Mystrioti et al.
Figure 1. Add curves to the figure.
Figure 2. Authors must sign all peaks.
Magnetic properties:
- There is no “table” with ultra-fine parameters
- Comparing the spectra obtained at different temperatures is a strange exercise in principle (even visually, it can be seen that the fields differ greatly at different temperatures). The spectra of “magnetite” and "maghemite" change strongly with temperature, in addition, the shape of the spectra is significantly affected by the size factor, therefore, the spectra should be compared only at the same temperatures, but better at several values, so that the dynamics of the change in the spectra with temperature can be seen;
- Remove and bring MS only at the temperature of the expected phase transition (Verwey transition) is generally not correct in principle, because in this state, a "mixed"/transitional state of the substance is observed, the f-factor decreases greatly, which does not allow in principle to estimate the "content" of the subspectra. What did the authors want to show these?
- One should compare not the intensities of the resonance lines, but the areas (МII);
- The quality of the model description, at least of the second spectrum, is very poor (MIII) - see the envelope line and the experimental one at the edges of the spectrum (from -11 to -8 and from 9.5 to 11 mm/s) - they do not match. If the authors had given the difference spectrum (as expected), then this fact would have been clearly visible.
- The presence of the third subspectrum for the sample (MIII) is associated with the manifestation of the phenomenon of superparamagnetism in the substance [DOI: 10.1134/S1063774520030244] and/or relaxation processes [DOI: 10.1016/j .ceramint.2022.03.070], and in no case by the presence of a “second phase”. Perhaps the authors formed a solid solution of magnetite-maghemite, Fe3-dO4, where d can be determined from the experimental spectrum in a similar way [DOI: 10.1016/j.matchemphys.2019.04.022].
Thermal analysis:
Add the chemical reactions of phase transformation.
Table 2. Why didn't use Sips equation?
The conclusions should be more detailed. Authors should describe the main findings of the study by adding numbers and values, not just text. Use the division into paragraphs: 1) 2) 3) etc.
Round 2
Reviewer 1 Report
Lines 180-184 "The magnetic iron nanoparticles can be synthesized using iron from the pregnant leaching solution (PLS) derived from several hydrometallurgical processes, such as the electrolytic zinc process or the hydrometallurgical treatment of low grade laterites. These solutions contain very high concentrations of iron, which must be separated from the metals of commercial value, e.g. Ni, Zn, etc. " This is a proposal of the authors? In this case, I consider that this paragraph might be placed at the end of the Introduction before the objective of the manuscript, for example close to paragraph of lines 303-304
Line 292. Elemental iron is Fe(0). It is better to put (0) instead Fe0 that seems iron oxide.
At the end of the Introduction, before the novelty, authors need to include the objectives of the manuscript
Lines 376-377. Authors need to include one or more references after next sentence "In the majority of published studies, the precipitation of nanomagnetite from chloride solutions is carried out by mixing a ferrous chloride and a ferric chloride salt at the required molar ratio 1 to 2".
Materials: Authors need to included all materials used, for example, ammonia and the corresponding purity.
In general, methodology can be improved. For example, equipment and conditions used for ultrasound or for stripping. It recommend to perform a detailed description of experimental processes.
References "Pankratov et al. (2019, 2020)", author need to perform references with numbers in [ ].
Figure 5. In order to compare both figures, It would be better to use the same scale on the y-axis
Units: Second is s, not sec. Please correct it. Line 483, Correct ºC
Figure 1. How was determined the Fe(III) and Fe(II) in the aqueous solution?
Discussion of Point of zero charge can be improved.
Author Response
Dear Reviewer 1,
Please find attached the author's reply.
Best Regard

Reviewer 2 Report
The manuscript can be accepted.
Author Response
Dear Reviewer 2,
Please find attached the author's reply.
Best Regard

Reviewer 3 Report
The authors answered all the questions in detail.
The Introduction section now has many links to the most recent research.
The magnetic properties' section has been significantly improved.
In this form, the article can be accepted in Minerals.
Author Response
Dear Reviewer 3,
Please find attached the author's reply.
Best Regard
